# RMNet: Equivalently Removing Residual Connection from Networks

## Abstract

Although residual connection enables training very deep neural networks, it is not friendly for online inference due to its multi-branch topology. This encourages many researchers to work on designing DNNs without residual connections at inference. For example, RepVGG re-parameterizes multi-branch topology to a VGG-like (single-branch) model when deploying, showing great performance when the network is relatively shallow. However, RepVGG can not transform ResNet to VGG equivalently because re-parameterizing methods can only be applied to linear Blocks and the non-linear layers (ReLU) have to be put outside of the residual connection which results in limited representation ability, especially for deeper networks. In this paper, we aim to remedy this problem and propose to remove the residual connection in a vanilla ResNet equivalently by a reserving and merging (RM) operation on ResBlock. Specifically, the RM operation allows input feature maps to pass through the block while reserving their information and merges all the information at the end of each block, which can remove residual connections without changing the original output. RMNet basically has two advantages: 1) it achieves a better accuracy-speed trade-off compared with ResNet and RepVGG; 2) its implementation makes it naturally friendly for high ratio network pruning. Extensive experiments are performed to verify the effectiveness of RMNet. We believe the ideology of RMNet can inspire many insights on model design for the community in the future.

## 1 Introduction

Since AlexNet (Krizhevsky et al., 2012) came out, the state-of-the-art CNN architecture has become deeper and deeper. For example, AlexNet only has 5 convolutional layers, it is soon extended to 19 and 22 layers by VGG network (Simonyan & Zisserman, 2015) and GoogLeNet(Szegedy et al., 2015; 2016; 2017), respectively. However, deep networks that simply stack layers are hard to train because of the gradient vanishing and exploding problem — as the gradient is back-propagated to earlier layers, repeated multiplication may make the gradient infinitely small or large. This problem has been largely addressed by the normalized initialization (LeCun et al., 2012; Glorot & Bengio, 2010; Saxe et al., 2013; He et al., 2015) and intermediate normalization layers (Ioffe & Szegedy, 2015), which enable networks with tens of layers converging. Meanwhile, another degradation problem has been exposed: with the increase of the network depth, accuracy gets saturated and then degrades rapidly. ResNet (He et al., 2016a;b) addresses the degradation problem and achieves 1K+ layers models by adding a residual connection from the input of a block to the output. Instead of hoping each stacked layer directly fit a desired underlying mapping, ResNet lets these layers fit a residual mapping. When the identity mapping is optimal, it would be easier to push the residual to zero than to fit an identity mapping by a stack of nonlinear layers. As ResNet gains more and more popularity, researchers propose many new architectures, e.g. (Hu et al., 2018; Li et al., 2019; Tan & Le, 2019; 2021), based on ResNet and interpret the success from different aspects.

However, ResDistill (Li et al., 2020) points that the residual connections in ResNet-50 account for about 40 percent of the entire memory usage on feature maps, which will slow down the inference process. Besides, residual connections in the network are not friendly for 'network pruning' (Ding et al., 2021b). By contrast, the VGG-like model, also called the plain model in this paper, has only one path and is fast, memory economical, and parallelization friendly. RepVGG (Ding et al., 2021b) proposes a powerful approach to remove residual connections via re-parametrization at inference.

Specifically, RepVGG adds 3×3 convolution, 1×1 convolution, and identity together for training. The BN layer was added at the end of each branch and the ReLU was added after the addition operation. During training, RepVGG only needs to learn the residual mapping, while in the inference stage, re-parameterization was utilized to transform the basic block of RepVGG into a stack of $3 \times 3$ convolution layers plus ReLU operation, which has a favorable speed-accuracy trade-off compared to ResNet. However, we find that the performance of RepVGG suffers severe degradation when the network goes deeper, we provide experiments to verify this observation in Section 4.1.

In this paper, we introduce a novel approach named RM operation, which can remove the residual connection with non-linear layers inside it and keep the result of the model unchanged. RM operation reserves the input feature maps by the first convolution layer, BN Layer and ReLU Layer, then merges them with the output feature maps by the last convolution in the ResBlock. With this method, we can equivalently convert a pre-trained ResNet or MobileNetV2 to an RMNet model to increase the degree of parallelism. Besides, the architecture of RMNet makes it friendly for pruning since there are no residual connections. We summarized our main contributions as follows:

- We find that removing residual connection by re-parameterization methods has its limitation, especially when the depth of the model is large. It lies in the non-linear operation that can not be put inside of residual connection for re-parameterization.
- We propose a novel method named RM operation which can remove residual connection across non-linear layers without changing the output by reserving the input feature maps and merging them with the output feature maps.
- With RM operation, we can convert the ResBlocks to a stack of convolutions and ReLUs, which achieves a better accuracy-speed trade-off and make it very friendly for pruning.

## 2 RELATED WORK

**Advantage of Residual Networks over Plain Model.** He et al. (He et al., 2016a) introduce ResNet for addressing the degradation problem. In addition, the gradient of layers in PreActResNet (He et al., 2016b) does not vanish even when the weights are arbitrarily small, which leads to nice backward propagation properties. (Weinan, 2017) shows the connection between dynamical systems and deep learning. (Lu et al., 2018) regards ResNet as an Euler discretization of Ordinary Differential Equations (ODE). NODEs (Chen et al., 2018) replaces residual block with neural ODEs for better training extremely deep networks. (Balduzzi et al., 2017) identifies shattered gradients problems making the training of DNNs difficult. They show that the correlation between gradients in the standard feedforward network decays exponentially with depth. In contrast, the gradients in ResNet are far more resistant to shattering, decaying sublinearly. (Veit et al., 2016) shows that residual networks can be seen as a collection of many paths of differing lengths. From this view, $n$-blocks' ResNet have $O(2^n)$ implicit paths connecting input and output, and that adding a block doubles the number of paths.

**DNNs Without Residual Connection.** A recent work(Oyedotun et al., 2020) combined several techniques including Leaky ReLU, max-norm, and careful initialization to train a 30 layer plain ConvNet, which could reach 74.6% top-1 accuracy, 2% lower than their baseline. A mean-field theory-based method (Xiao et al., 2018) is proposed to train extremely deep ConvNets without branches. However, 1K-layer plain networks only achieve 82% accuracy on CIFAR-10. (Balduzzi et al., 2017) use a "looks linear" initialization method and CReLUs (Shang et al., 2016) to train a 198 layer plain model to 85% accuracy on CIFAR-10. Though the theoretical contributions were insightful, the models were not practical. One can get DNNs without residual connection in a re-parameterization manner: re-parameterization (Ding et al., 2019; 2021a) means using the params of a structure to parameterize another set of params. Those approaches first train a model with residual connection and remove residual connection by re-parameterization when inference. DiracNet (Zagoruyko & Komodakis, 2017) uses the addition of an identity matrix and convolutional matrix for forwarding propagation, the parameters of convolution need only to learn the residual function like ResNet. After training, DiracNet adds the identity matrix to the convolutional matrix and uses the reparameterized model for inference. However, DiracNet can only train up to 34 layer plain model which has 72.83% accuracy on ImageNet. RepVGG (Ding et al., 2021b) deploys residual neural network only at the training time. At the inference, RepVGG can transform the residual block into a plain module consisting of a stack of $3 \times 3$ convolution and ReLU via re-parameterization.

The difference between DiracNet and RepVGG is each block in DiracNet has two branches (identity without BN and $3 \times 3$ ConvBN) while RepVGG has three branches (identity with BN, $3 \times 3$ ConvBN, and $1 \times 1$ ConvBN). However, those re-parameterization methods that used commutative property can only apply to linear layers, i.e., the non-linear layers have to be outside of the residual connection, which limits the potential of neural networks for large depths.

**Filter Pruning:** Filter pruning is a promising solution to accelerate CNNs. Numerous inspiring works prune the filters by evaluating their importance. Heuristic metrics are proposed, such as the magnitude of convolution kernels (Li et al., 2017), the average percentage of zero activations (APoZ) (Hu et al., 2016). There also exists some work to let networks automatically select important filters. For example, (Liu et al., 2017a) sparsify the weights in the BN layer to automatically find which filters contribute most to the network performance. However, For ResNet-based architecture, the existence of residual connection limits the power of pruning since the dimension of the input and output through the residual connection must keep the same. Thus pruning ratio of ResNet is not larger than the ratio of the plain model. Since RM operation can equivalently transform ResNet to a plain model, the transferred model (RMNet) also has a great advantage on pruning.

## 3 RM OPERATION AND RMNET: TOWARDS AN EFFICIENT PLAIN NETWORK

### 3.1 RM OPERATION

Figure 1 shows the process of equivalently removing residual connection by RM Operation. For simplicity, we do not show BN layer in the figure, and the number of input channels, medium channels, and output channels are the same and are assigned as $C$.

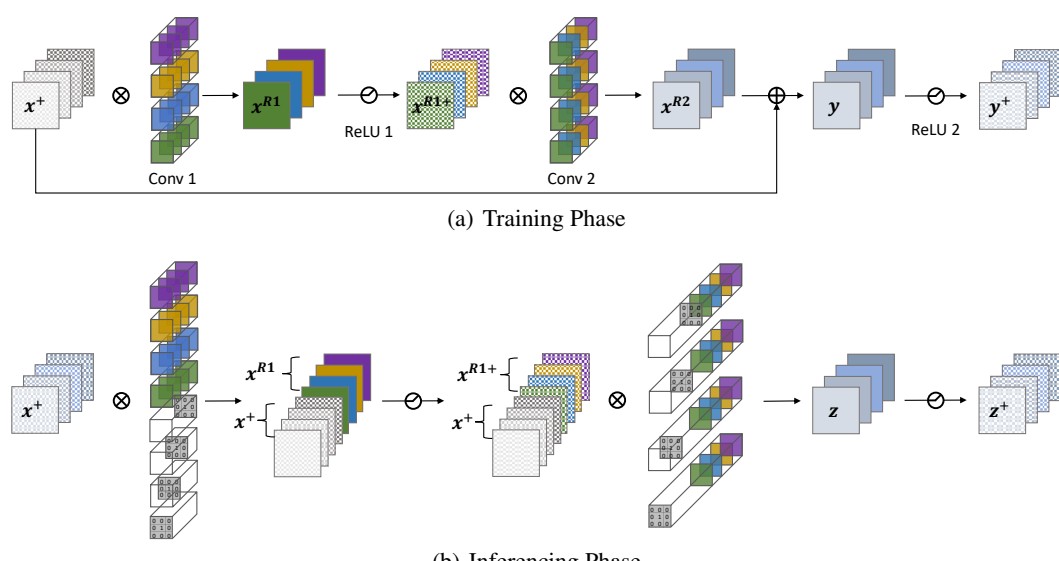

(a) Training Phase

(b) Inferencing Phase

Figure 1: The upper figure shows a ResBlock during the training phase. The lower figure is the converted RMBlock for inference, which has no residual connections. Both blocks have equal output given the same input.

● **Reserving:** We first insert several Dirac initialized filters (output channels) in Conv 1. The number of Dirac initialized filters is the same as the input channels' number in the convolution layer. The Dirac initialized filters are defined as 4-dimensional matrix:

$$I_{c,n,i,j} = \begin{cases} 1 & if\ c = n\ and\ i = j = 0, \\ 0 & otherwise \end{cases} \tag{1}$$

We can view these filters in Figure 1(b). Every filter only has one element to be 1, which can reserve the corresponding channel's information of the input feature map via convolution.

For the BN layer, to reserve the input feature map, weight $w$ and bias $b$ in BN need to be adjusted to make the BN layer behave like an identity function. Suppose the running mean and the running variance of the feature maps are $\mu$ and $\sigma^2$ separately, we set $w = \sqrt{\sigma^2 + \epsilon}$ and bias $b = \mu$. Then for any input $x$ through the BN layer, the output is:

$$
\begin{aligned}
y &= w \times \frac{(x - \mu)}{\sqrt{\sigma^2 + \epsilon}} + b \\
&= x
\end{aligned}
\tag{2}
$$

where $\epsilon = 10^{-5}$ is added to the $\sigma^2$ in case of the divisor to be zero.

For the ReLU layer, There are two cases to be considered:

- When the input value through residual connection is non-negative (*ie.e,* in ResNet, every ResBlock has a following a ReLU layer, which keeps input values are all non-negative), we can directly use ReLU to reserve the information. Since ReLU does not change the non-negative values.

- When the input value through residual connection can be negative (e.g. in MobileNetV2, ReLU is only located in the middle of the ResBlock), we use PReLU instead of ReLU to reserve the information. The parameters of PReLU regarding to the additional channels are set to one. In this way, PReLU behaves as Identity function.

From the above analysis, the input feature map can be well reserved by Conv 1, BN, and ReLU in the ResBlock.

• **Merging:** We extend input channels in Conv 2 and dirac initialize these channels. The value of $i$-th channel of $z$ is the sum of the original i-th filters' output $x_i^{R2}$ and the i-th input feature map $x_i^+$, which is equal to $y^i$ in the original ResBlock. The whole merging process can be formulated as:

$$
\begin{aligned}
z_{c,h,w}^+ &= Max(\sum_{n}^{2C} \sum_{i}^{K} \sum_{j}^{K} \left([W^{R2}, I]_{c,n,i,j} \times [x^{R1+}, x^+]_{n,h+i,w+j} + [B^{R2}, 0]_c\right), 0) \\
&= Max(\sum_{n}^{C} \sum_{i}^{K} \sum_{j}^{K} \left(W_{c,n,i,j}^{R2} \times x_{n,h+i,w+j}^{R1+} + B_c^{R2}\right) \\
&\quad + \sum_{n=C+1}^{2C} \sum_{i}^{K} \sum_{j}^{K} \left(I_{c,n,i,j} \times x_{n,h+i,w+j}^+ + 0\right), 0) \\
&= Max(\sum_{n}^{C} \sum_{i}^{K} \sum_{j}^{K} \left(W_{c,n,i,j}^{R2} \times x_{n,h+i,w+j}^{R1+} + B_c^{R2}\right) + x_{c,h,w}^+, 0) \\
&= {y^+}_{c,h,w}
\end{aligned}
\tag{3}
$$

where $[*, *]$ means the concatenation of the two elements.

Thus, by reserving and merging, we can remove the residual connection without changing the original output of ResBlock.

## 3.2 CONVERT RESNET TO VGG

In this section, we show how to apply RM operation to convert ResNet into VGG-like RMNet. Notice that there exist some DownSample ResBlocks at the first block of layer2, layer3, and layer4 in ResNet. The function of 'downsample' is for:

- doubling the number of channels;
- reducing the depth and width of the feature map by half.

Even though there are only a few DownSample ResBlocks in ResNet, removing residual connection in such blocks is not as direct as in basic ResBlocks because there exist $1 \times 1$ convolution in the

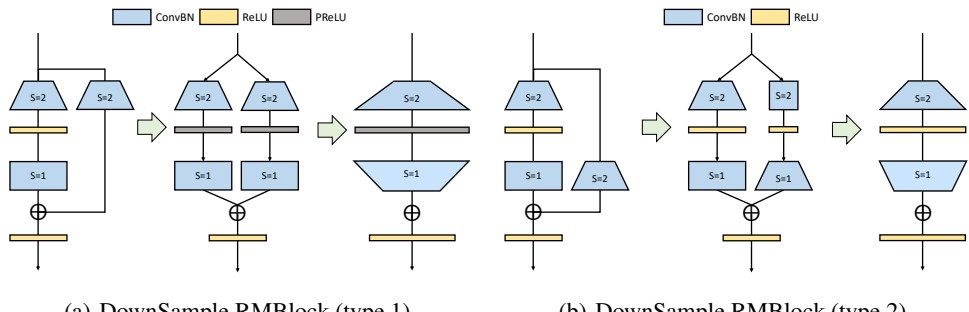

(a) DownSample RMBlock (type 1)       (b) DownSample RMBlock (type 2)

Figure 2: (a) and (b) show two methods of removing downsample branch in the original Down-Sample ResBlock.

downsample branch and it will change the input value in the path of residual connection. We propose two solutions in Figure 2 to convert DownSample ResBlock to RMBlock.

As shown in Figure 2(a), the original 1×1 downsample kernel is padding with zeros to 3×3 kernel. we replace ReLU in the block with PReLU, whose parameters for each channel can control the scale of negative activation, as analyzed in Section 3.1. For the left (residual) branch, we set the PReLU parameters as zeros, in this way, the function of the PReLU is equal to ReLU. PReLU is added after the downsample convolution, the parameters of which are initialized by one. In this way, PReLU is equal to the Identity Mapping function. New additional channels of the second convolutional layer are Dirac initialized (similar to Figure 1(b)), which keep the feature map after downsampling convolution unchanged. Then, all the layers in the left and right branches can be merged together into one sequence shown in Figure 2(a).

In Figure 2(b), we separate the function of downsampling convolution into two convolutions in downsample branch. The first module is 3×3 convolution with Dirac initialized filters and the stride is 2 for reducing the depth and width of the input feature map by half. The feature map through this Conv layer is still non-negative and ReLU can reserve the value unchanged. The second module is 3×3 convolution, whose weights are transferred from original $1 \times 1$ filters in downsample branch with 0 paddings. Since the original filters' kernel size is 1×1, reducing the size of feature maps before doubling the number of channels will not change the receptive field in the original downsample branch.

Both methods can equally convert the DownSample Blocks to RMBlocks. The number of parameters of the first method is $108C^2 + 4C$, including Conv 1 ($C \times 4C \times 3 \times 3$), PReLU ($4C$), Conv 2 ($4C \times 2C \times 3 \times 3$). The number of parameters of the second method is $81C^2$, consisting of Conv 1 ($C \times 3C \times 3 \times 3$), Conv 2 ($3C \times 2C \times 3 \times 3$). We can see the number of parameters of the second method is only 0.75 times of the first method. Thus we use the second method to convert ResNet to RMNet in our experiment.

### 3.3 CONVERT MOBILENETV2 TO MOBILENETV1

Technically, there is no difference between converting ResNet and converting MobileNetV2 by using RM operation. However, the specialty of the structure of MobileNetV2 allows to further decrease the inference time by using **parameter fusion** after RM operation.

From Figure 3, after applying RM operation on MobileNetV2, the residual connections are removed, which leaves two **Pointwise-ConvBN** layers shown in the dashed box. Since convolution and batch-normalization can be both presented by matrix multiplication and there exists no non-linear layer between them, these two Pointwise-ConvBN layers can be fused. Mathematically, let $X$, $Y$, $W$, $B$ be the input feature map, output feature map, weight, and bias of the first Pointwise-ConvBN layer in the dashed box and $Z$, $W^{'}$, $B^{'}$ be the output feature map, weight and bias of the second Pointwise-ConvBN layer in the dashed box. The process of fusing the parameters can be represented as:

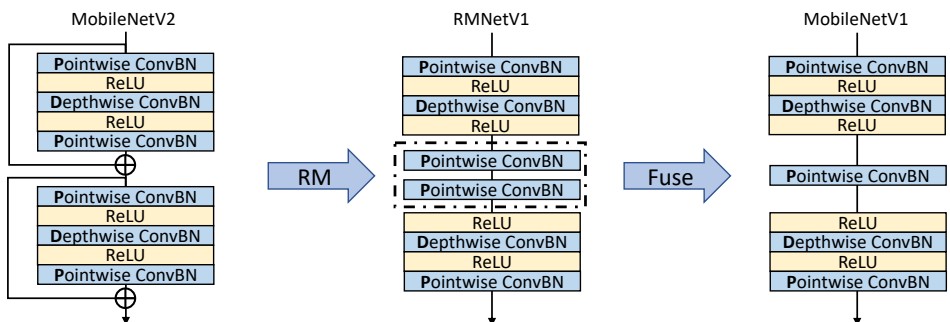

Figure 3: The process of converting MobileNetV2 into MobileNetV1.

$$
\begin{aligned}
Z &= W^{'}Y + B^{'} \\
&= W^{'}(WX + B) + B^{'} \\
&= (W^{'}W)X + W^{'}B + B^{'}
\end{aligned}
\tag{4}
$$

The fused weight is $W^{'}W \in R^{n,c,1,1}$ and the fused bias is $W^{'}B + B^{'} \in R^{n}$. Thus, RM operation provides another opportunity to further decrease the inference time by fusing the parameters. After parameter fusion, the architecture of RMNet is identical to MobileNetV1 which is very interesting since the presence of MobileNetV2 is to increases the generalization ability of MobileNetV1 by utilizing residual connections. However, we show RM operation can inverse this process, *i.e.,* converting MobileNetV2 into MobileNetV1 to make MobileNetV1 great again. We provide experiments in Section 4.3 to verify the effectiveness of this transformation.

### 3.4 PRUNING RMNET

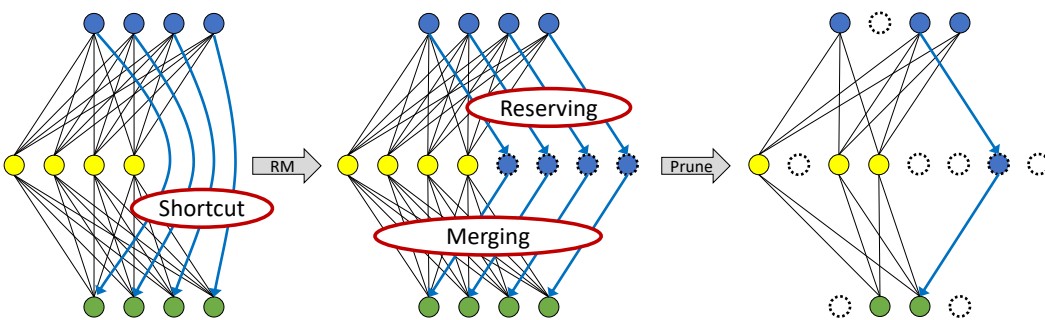

Figure 4: The process of pruning on ResNet. During the process, RMNet can serve as a transition to gain larger pruning ratio.

Since RMNet does not have any residual connections, it is more friendly for filter pruning. In this paper, we adopt Network slimming (Liu et al., 2017b) to prune RMNet because of its simpleness and effectiveness. Specifically, we first train ResNet and sparsity the weight in the BN layer. Note an additional BN layer should be added in the residual connection during training since we also need to determine which additional filters are important after RM operation. After training, we convert ResNet into RMNet and prune the filters according to the weights in BN layers, which is identical to vanilla Network slimming. Figure 4 shows the overall process of pruning. Different from the traditional approach, RMNet can serve as a transition to gain a larger pruning ratio.

## 4 EXPERIMENTS

This section aranges as follows: in Section 4.1, we show RMNet has great advantage over RepVGG for deeper networks; in Section 4.2. we show RMNet can achieve better speed-accuracy tradeoff on CIFAR10, CIFAR100 and ImageNet datasets; in Section 4.3, we show RMNet is applicable for light-weight models; in Section 4.4, we verify the effectivenss of RMNet on network pruning task.

The speed of each model is tested on a Tesla V100 GPU with a batch size of 128 by first feeding 50 batches to warm the hardware up, then 50 batches with time usage recorded.

### 4.1 REMAINING HIGH ACCURACY FOR DEEPER NETWORK

One power of DNN is that the representation ability increases as we raise the network depth. Thus we examine how network depth influences network performance on RMNet and RepVGG. Figure 5 shows the network performance on CIFAR-10 and CIFAR-100. For a fair comparison, the same hyper-parameters are deployed for each method: mini-batch size (256), optimizer (SGD), initial learning rate (0.1), momentum (0.9), weight decay (0.0005), number of epochs (200), learning rate decay 0.1 at every 60 epochs.

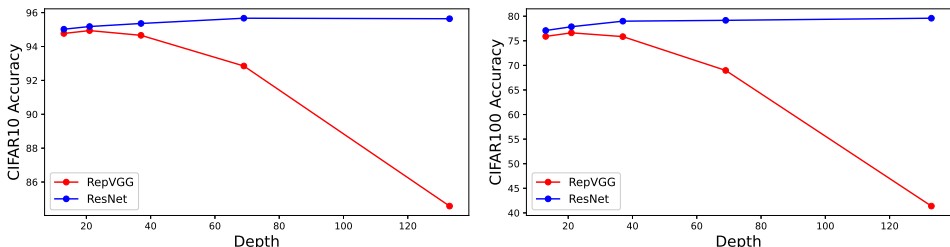

Figure 5: This figure shows the influence of depth on RMNet and RepVGG.

From Figure 5, as the depth increases, RMNet can get better accuracy. This is because, unlike RepVGG, RMNet can **equivalently** convert ResNet into a plain model. Thus the performance will not decrease as the network goes deeper. In contrast, the accuracy of RepVGG will decrease (the reasons are analyzed at Appendix A.2 ), e.g. compared to RMNet 133 that achieves 79.57% accuracy on CIFAR-100, RepVGG 133 only has 41.38% accuracy.

### 4.2 BETTER ACCURACY-SPEED TRADEOFF

From Section 3, RMNet removes residual connections in the cost of bringing additional parameters. For example, in Figure 1, RM operation doubles the number of parameters of the original ResBlock. To alleviate this issue, we use RM operation on the ResNet-based architectures with **Inverted Residual Block** for designing RMNet. Inverted Residual Block (IRB) mainly contains three consecutive steps: point-wise convolution, group-wise convolution, and point-wise convolution. The first 'point-wise convolution' will increase the number of channels of the input feature map by $T$ times and the second 'point-wise convolution' will reduce the number of channels to the original number. For point-wise convolution, RM operation only increases the number of parameters and FLOPs by $\frac{1}{T}$ times. Suppose the size of the input feature map is $H \times W \times C$ and the parameters of the first point-wise convolutional layer are expressed as $T \times C \times C \times 1 \times 1$. We only need additional $C \times C \times 1 \times 1$ parameters to reserve the input feature map, since RM operation is only to reserve the input feature map. 'Group-wise convolution' is also parameter-friendly. For example, for a normal convolution, we need $C \times K \times K$ parameters to reserve the information of an input channel. While for group-wise convolution, we only need $\frac{1}{G}$ times of parameters to reserve the information where $G$ is the group number in group-wise convolution. Besides, the cost of storage and calculation does not change whether or not to use group-wise convolution. Based on the advantage of IRB, we design a series of RMNet models. We first determine the input and output width by the classic width setting of [64, 128, 256, 512]. We replace $7 \times 7$ Conv and MaxPooling on the head with two sequences of $3 \times 3$ Conv, BN, and ReLU (the same approach used in RepVGG). The tail of RMNet is the same as

ResNet. For RMNet 50x6_32, '50' indicates the number of all the Conv layers; '6' indicates the multiple of classic width setting, for example, the width of the grouped convolution layer in RMNet 50x6_32 is [6×64, 6×128, 6×256, 6×512]; '32' indicates the number of channels of each group in RMBlock. We print out the detailed architecture of RMNet in the Appendix A.4. Next, we compare RMNet with SOTA models on CIFAR10, CIFAR100 and ImageNet to test the speed and accuracy. For a fair comparison, we train **RMNet** following the official implementations of RepVGG [1].

Table 1: Comparing RMNet with other SOTA models. RMNet 26 have [2, 2, 2, 2] RMBlocks for each stage; RMNet 41 have [2, 3, 5, 3] RMBlocks for each stage; RMNet 50 have [3, 4, 6, 3] RMBlocks for each stage; RMNet 101 have [3, 4, 23, 3] RMBlocks for each stage; RMNet 152 have [3, 8, 36, 3] RMBlocks for each stage.

| Dataset | Backbone | Params(M) | FLOPS(G) | Accuracy(%) | Imgs/sec |
|---------|----------|-----------|----------|-------------|----------|
| CIFAR10 | ResNet 34 | 21.2 | 1.16 | 95.64 | 8208.4 |
|  | DiracNet 34 | 21.1 | 1.15 | 94.83 | 6299.64 |
|  | ResDistill 34 | 21.1 | 1.16 | 94.62 | 9652.8 |
|  | RepVGG A2 | 24.1 | 1.67 | 95.35 | 8099.7 |
|  | **RMNet 26×3_32** | **8.8** | **0.51** | **95.81** | **10078.2** |
| CIFAR100 | ResNet 34 | 21.3 | 1.16 | 78.61 | 8205.2 |
|  | DiracNet 34 | 21.1 | 1.15 | 76.04 | 6285.3 |
|  | ResDistill 34 | 21.2 | 1.16 | 78.42 | 9621.9 |
|  | RepVGG A2 | 24.2 | 1.67 | 78.41 | 8084.3 |
|  | **RMNet 26×3_32** | **8.9** | **0.51** | **79.16** | **10065.5** |
| ImageNet | ResNet 50 | 25.6 | 4.11 | 76.31 | 1252.1 |
|  | ResDistill 50 | 25.6 | 4.11 | 76.08 | **1494.4** |
|  | **RMNet41×4_8** | **11.2** | **1.9** | **77.80** | 1460.4 |
|  | ResNeXt 50 | 25.0 | 4.26 | 78.41 | 916.9 |
|  | RepVGG B1 | 51.8 | 11.82 | 78.31 | 1123.2 |
|  | **RMNet 41×5_16** | **23.9** | **3.79** | **78.5** | **1185.8** |
|  | ResNet 101 | 44.5 | 7.83 | 77.21 | 729.9 |
|  | VGG 16 | 138 | 15.48 | 72.21 | 782.1 |
|  | RepVGG B2 | 80.3 | 18.38 | 78.78 | 712.9 |
|  | **RMNet 50×5_32** | **39.6** | **6.88** | **79.08** | **849.8** |
|  | ResNet 152 | 60.19 | 11.56 | 77.78 | 512.8 |
|  | RepVGG B3 | 111.0 | 26.21 | 79.26 | 543.5 |
|  | **RMNet 50×6_32** | **47.6** | **8.26** | **79.57** | **713.8** |
|  | ResNeXt 101 | 88.8 | 16.48 | 79.88 | 308.4 |
|  | **RMNet 101×6_16** | **59.45** | **11.11** | **80.07** | **461.5** |
|  | **RMNet 152×6_32** | **126.92** | **25.32** | **80.36** | **272.9** |

From Table 1, the accuracy of RMNet 50×6_32 is 2.3% higher than ResNet 101 and 0.59% higher than RepVGG B2 with the nearly same speed. It is worth noting that RMNet 101×6_16 reaches over 80% top-1 accuracy without using any tricks, which is the first time for a plain model, to the best of our knowledge. Also, increasing the depth to 152 still has a benefit to RMNet, which shows the great potential of our method.

## 4.3 FRIENDLY FOR LIGHT-WEIGHT MODELS

We conduct an experiment to verify our analysis in Section 3.3. We first train a MobileNetV2 from scratch and convert it to RMNetV1 following the process introduced in Section 3.3. Note that the initial architecture of MobileNetV2 has to be designed to make sure the transformed RMNetV1 has the same architecture (depth and width) of MobileNetV1 from the original paper (Howard et al., 2017). Thus the MobileNetV2 in Figure 3 is different from vanilla MobileNetV2(Sandler et al.,

---
[1]https://github.com/DingXiaoH/RepVGG/blob/main/train.py

2018). We also train a MobileNetV1 trained from scratch for comparison. The result are show in Table 2.

Table 2: Converting MobileNetV2 to MobileNetV1 by RM operation and Fuse Operation on CIFAR-10 and CIFAR100.

| Dataset | Backbone | Initial | w/wo Training | FLOPS(M) | Imgs/Sec | Acc(%) |
|---------|----------|---------|---------------|----------|----------|--------|
| CIFAR-10 | MobileNetV2 | Scratch | ✓ | 83.4 | 19119 | **92.07±0.18** |
| | RMNetV1 | MobileNetV2 | × | **47.1** | **27720** | **92.07±0.18** |
| | MobileNetV1 | Scratch | ✓ | **47.1** | **27720** | 91.31±0.11 |
| CIFAR-100 | MobileNetV2 | Scratch | ✓ | 83.5 | 19076 | **70.57±0.09** |
| | RMNetV1 | MobileNetV2 | × | **47.2** | **27712** | **70.57±0.09** |
| | MobileNetV1 | Scratch | ✓ | **47.2** | **27712** | 68.75±0.37 |

From the experiment in Table 2, we can see the speed of RMNetV1 (equivalently converted from MobileNetV2) is faster than vanilla MobileNetV2 and the performance of RMNetV1 is higher than MobileNetV1, which shows RM operation can be very friendly for light-weight models.

### 4.4 FRIENDLY FOR PRUNING

We conduct an experiment in Figure 6 to verify the effectiveness of pruning RMNet. We use $L_1$ norm multiply a certain sparsity factor to punish the weight of BN layers, and regard the channels whose weight is smaller than a certain threshold as invalid. The sparsity factor is selected from 1e-4 to 1e-3, and the threshold is selected from 5e-4 to 5e-3. A larger degree of sparsity will lead to a larger pruning ratio.

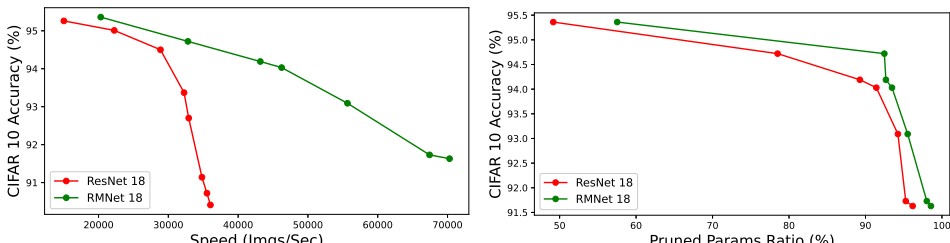

Figure 6: (a) and (b) show the CIFAR10 accuracy with respect to network speed and pruned parameters ratio in pruning task.

From the experiments, RMNet retains higher accuracy than ResNet under a larger pruning ratio and when the speed of pruned RMNet is nearly the same with pruned ResNet, because of more reasonable structure. Thus RMNet has a better accuracy-speed tradeoff over ResNet architecture on pruning tasks.

## 5 CONCLUSION AND DISCUSSION

In this paper, We propose RM operation to remove residual connections and perform RM operation to convert ResNet and MobileNetV2 into plain models (RMNet). RM operation allows input feature maps to pass through the block while reserving their information and merges all the information at the end of each block, which can remove residual connection without changing original output. Experiments have shown that RMNet can achieve a better speed-accuracy trade-off and is very friendly for network pruning.

There are some future directions to be considered: 1) Analyzing the residual connection on the Transformer based architectures and removing such connections via RM operation. 2) Searching residual networks with less residual connection with NAS and converting them to efficient RMNets.

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

# A APPENDIX

The appendix is arranged as follows: in Section A.1, we perform ablation studies; in Section A.2, we further analyze why RepVGG does not have high performace for deeper network; in Section A.3, we show fine-tuning RMNet can achieve higher performance; in Section A.4, we show the detailed architecture of RMNet.

## A.1 ABLATION STUDY

In this section, we perform an ablation study to show how batch-size and hyper-parameters of network structure affect inference time for RM operation. The results are shown in Figure 7. As shown in the left bottom sub-figure, the exact inference speed grows linearly with batch size when the batch size is relatively small, indicating the resource are sufficient. From the experiments, we can find that under the condition of sufficient resources, *i.e.,* lower batch-size and lower complexity of network structure, RMNet has a great advantage over ResNet. This is because RMNet is a plain model without residual connections. The degree of parallelism can be greatly increased when the resources are sufficient. Thus we believe RMNet has great potential in the future.

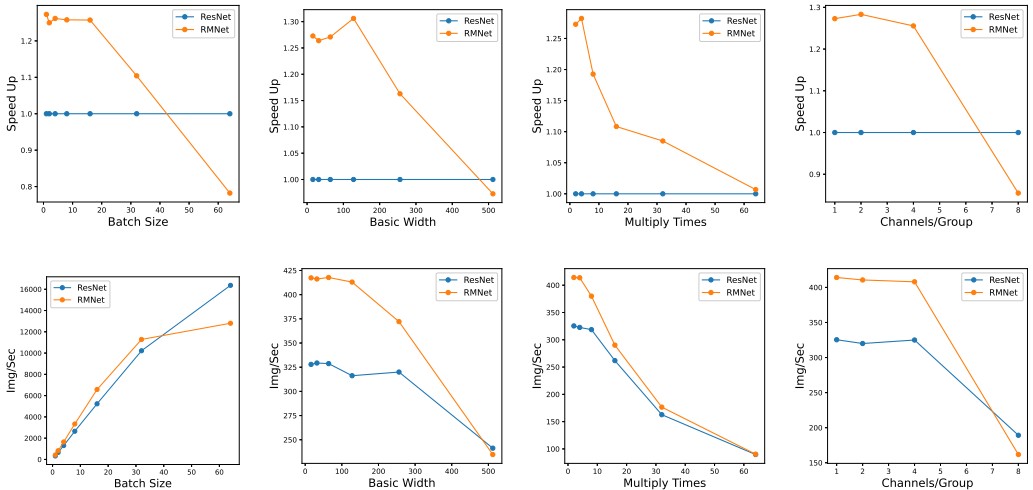

Figure 7: The first row shows the speed-up ratio of RMNet over ResNet. The second row shows the exact inference speed for ResNet and RMNet. Basic width, Multiply times and Channels/Groups are hyper-parameters of Network Structure introduced in Section 4.2. The larger the value, the more complex the network structure.

## A.2 COMPARISON OF REPVGG AND RESNET

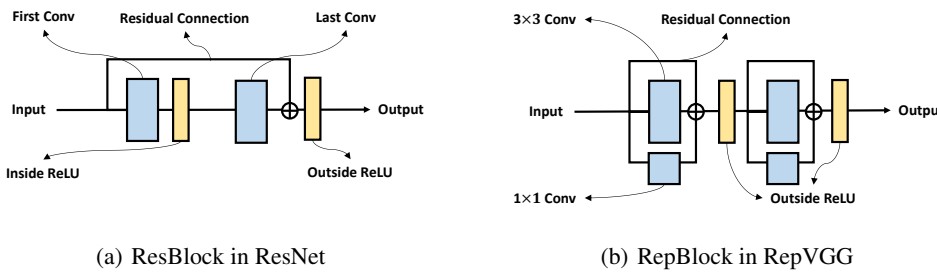

(a) ResBlock in ResNet        (b) RepBlock in RepVGG

Figure 8: Comparison of ResBlock in ResNet and RepBlock in RepVGG.

We depict the basic blocks used in ResNet and RepVGG in Figure 8. In ResBlock (Figure 8(a)), two ReLUs are inside and outside of the residual connection, respectively. While in RepBlock (Figure 8(b)), since re-parameterization is based on the commutative law of multiplication, two ReLUs have to be both outside of the residual connection. One can build RepVGG from a basic ResNet architecture by replacing each ResBlock with two RepBlocks.

Next, We analyze why RepVGG can not be trained very deep like ResNet from both forward and backward passes:

• **forward path:** (Veit et al., 2016) assumes that the success of ResNet can be contributed to the "model ensemble". we can regard ResNet as an ensemble of many paths of different lengths. Thus, $n$-blocks' ResNet have $O(2^n)$ implicit paths connecting input and output. However, unlike ResNet that two branches in the block are separable and can not be merged, multi-branches in RepVGG can be represented by one branch, which can be shown as:

$$
\begin{aligned}
x_{i+1} &= ReLU(BN(x_i) + BN(Conv_{1\times1}(x_i)) + BN(Conv_{3\times3}(x_i))) \\
&= ReLU(BN(Conv'_{3\times3}(x_i)))
\end{aligned}
\tag{5}
$$

where $Conv'$ is the merged convolution of each $Conv$ in the branches. Thus RepVGG does not have the implicit "ensemble assumption" of ResNet, and the representation gap between RepVGG and ResNet increases as the number of blocks raises.

• **backward path:** (Balduzzi et al., 2017) analyzes "shattered gradients problem" in deep neural networks. The correlation of gradients behaves like 'White Gaussian Noise' when there are more ReLUs in the backward path. Suppose ResNet and RepVGG both have $n$ layers. From Figure 8(a), Information in ResNet can pass through the residual without going through inside ReLU in each block. However, each ReLU in RepVGG is located in the main path. Thus the number of ReLUs for ResNet in the backward path is $\frac{n}{2}$, while the number of ReLUs for RepVGG in the path is $n$ which suggests that the gradients in ResNet are far more resistant to shattering when the depth is large, leading to better performance than RepVGG.

Table 3 shows the performance of RepVGG and ResNet on ImageNet dataset. The setting of this experiment is followed by the official implementation of RepVGG. We can see under the same network structure, ResNet-18 is 0.5% higher than RepVGG-18, and ResNet-34 is 0.8% higher than RepVGG in terms of top-1 accuracy. Thus, RepVGG increases the speed at the cost of losing representation ability.

Table 3: Comparison of ResNet and RepVGG with the same depth and width on ImageNet.

| Backbone | Params(M) | FLOPS(G) | Top-1 Acc(%) | Top-5 Acc(%) | Imgs/sec |
|---|---|---|---|---|---|
| ResNet 18 | 11.7 | 1.82 | **71.146** | **90.008** | 4548.59 |
| RepVGG 18 | 11.5 | 1.80 | 70.638 | 89.608 | 4833.36 |
| ResNet 34 | 21.8 | 3.67 | **74.442** | **91.89** | 2634.93 |
| RepVGG 34 | 21.6 | 3.65 | 73.67 | 91.564 | 2772.17 |

### A.3 FINETUNE

Compared to other approaches (Li et al., 2020; Ding et al., 2021b; Zagoruyko & Komodakis, 2017) that remove residual connections, RM operation can equivalently convert ResBlock to a plain module without re-training. However, if we fine-tune the pre-trained network, the performance may slightly improve since RM operation brings additional parameters into the network. It is worth noting that we need to carefully design the BN layer of additional channels (brought by RM operation) during fine-tuning since the running mean and running var in BN may lead to an unstable state of DNN. Thus We statistics the mean and variance by inputting the training dataset through the network, which will serve as the initial values of running mean and running var of BN layers for better stableness. In addition, the weight and bias in BN are set by Equation 2 to guarantee the equality of the model. Of course, there exists another solution that we do not add BN layers when fine-tuning, *i.e.,* fusing the BN with convolution before fine-tuning. However, it will lead to worse performance.

Figure 9 shows the accuracy of ResNet 18/34 on CIFAR-10/100 with/without fusing the BN layer when training. We can see the network without fusing BN has better performance and the accuracy is better than the pre-trained model.

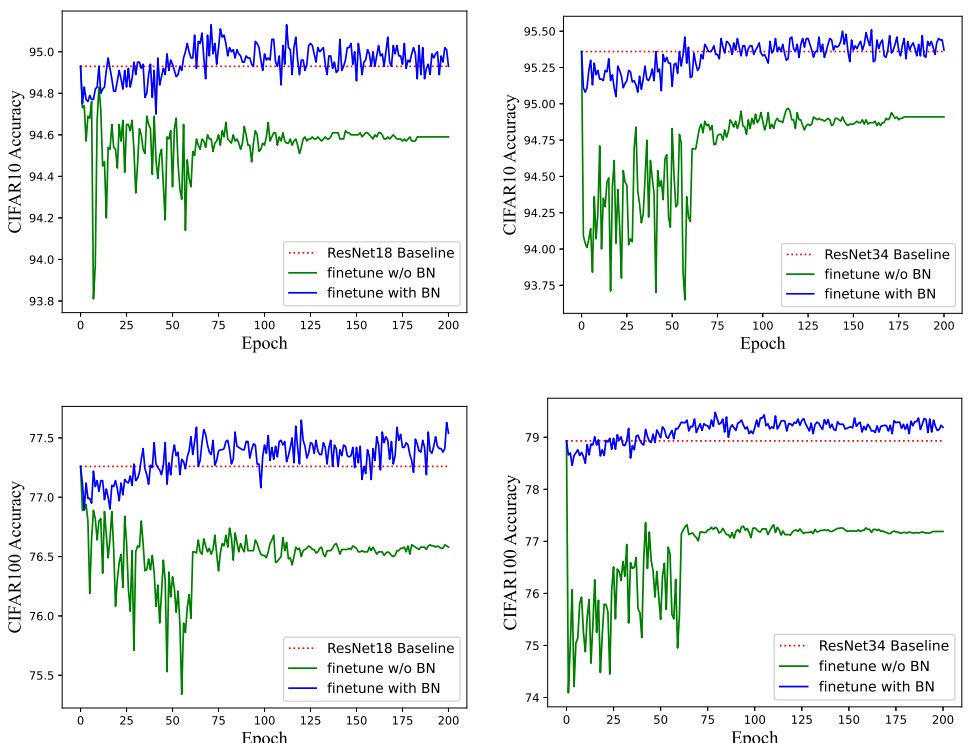

Figure 9: In each sub-figure, green line and blue line are the accuracies of the ResNet model with or without the BN layer, respectively.

### A.4 DETAILED STRUCTURE OF RMNET

We print out the detailed network structure of RMNet $26 \times 3\_32$ in Table 4. From Table 4, the most components in RMNet are 'Conv' and 'ReLU', making it very efficient on inference.

Table 4: The detailed network structure of RMNet $26 \times 3\_32$

```
Sequential(
(0): Conv2d(3, 64, kernel_size=(3, 3), stride=(1, 1), padding=(1, 1))
(1): ReLU(inplace=True)
(2): Conv2d(64, 192, kernel_size=(1, 1), stride=(1, 1))
(3): ReLU(inplace=True)
(4): Conv2d(192, 192, kernel_size=(3, 3), stride=(1, 1), padding=(1, 1), groups=6)
(5): ReLU(inplace=True)
(6): Conv2d(192, 64, kernel_size=(1, 1), stride=(1, 1))
(7): ReLU(inplace=True)
(8): Conv2d(64, 192, kernel_size=(1, 1), stride=(1, 1))
(9): ReLU(inplace=True)
(10): Conv2d(192, 192, kernel_size=(3, 3), stride=(1, 1), padding=(1, 1), groups=6)
(11): ReLU(inplace=True)
(12): Conv2d(192, 64, kernel_size=(1, 1), stride=(1, 1))
(13): ReLU(inplace=True)
(14): Conv2d(64, 320, kernel_size=(1, 1), stride=(1, 1))
(15): ReLU(inplace=True)
(16): Conv2d(320, 320, kernel_size=(3, 3), stride=(2, 2), padding=(1, 1), groups=5)
(17): ReLU(inplace=True)
(18): Conv2d(320, 128, kernel_size=(1, 1), stride=(1, 1))
(19): ReLU(inplace=True)
(20): Conv2d(128, 384, kernel_size=(1, 1), stride=(1, 1))
(21): ReLU(inplace=True)
(22): Conv2d(384, 384, kernel_size=(3, 3), stride=(1, 1), padding=(1, 1), groups=6)
(23): ReLU(inplace=True)
(24): Conv2d(384, 128, kernel_size=(1, 1), stride=(1, 1))
(25): ReLU(inplace=True)
(26): Conv2d(128, 640, kernel_size=(1, 1), stride=(1, 1))
(27): ReLU(inplace=True)
(28): Conv2d(640, 640, kernel_size=(3, 3), stride=(2, 2), padding=(1, 1), groups=5)
(29): ReLU(inplace=True)
(30): Conv2d(640, 256, kernel_size=(1, 1), stride=(1, 1))
(31): ReLU(inplace=True)
(32): Conv2d(256, 768, kernel_size=(1, 1), stride=(1, 1))
(33): ReLU(inplace=True)
(34): Conv2d(768, 768, kernel_size=(3, 3), stride=(1, 1), padding=(1, 1), groups=6)
(35): ReLU(inplace=True)
(36): Conv2d(768, 256, kernel_size=(1, 1), stride=(1, 1))
(37): ReLU(inplace=True)
(38): Conv2d(256, 1280, kernel_size=(1, 1), stride=(1, 1))
(39): ReLU(inplace=True)
(40): Conv2d(1280, 1280, kernel_size=(3, 3), stride=(2, 2), padding=(1, 1), groups=5)
(41): ReLU(inplace=True)
(42): Conv2d(1280, 512, kernel_size=(1, 1), stride=(1, 1))
(43): ReLU(inplace=True)
(44): Conv2d(512, 1536, kernel_size=(1, 1), stride=(1, 1))
(45): ReLU(inplace=True)
(46): Conv2d(1536, 1536, kernel_size=(3, 3), stride=(1, 1), padding=(1, 1), groups=6)
(47): ReLU(inplace=True)
(48): Conv2d(1536, 512, kernel_size=(1, 1), stride=(1, 1))
(49): ReLU(inplace=True)
(50): AdaptiveAvgPool2d(output_size=1)
(51): Flatten(start_dim=1, end_dim=-1)
(52): Linear(in_features=512, out_features=10, bias=True))
```

