# OpenReview forum: "RMNet: Equivalently Removing Residual Connection from Networks"
_ICLR.cc/2022/Conference — ICLR 2022 Submitted_

### Official Review · Reviewer_oEx8 · 2021-10-20

**Correctness:** 4
**Technical Novelty And Significance:** 3
**Empirical Novelty And Significance:** 4
**Recommendation:** 8
**Confidence:** 5

**Main Review:**

Strength:

1. The motivation of this paper is clear. Plain model can reduce off-chip memory access and avoid point operations like residual plus which is helpful for model deployment on specific platforms. This paper is also well-written and easy to follow.

2. The method of re-parameterization is very useful in model design and network pruning. However, the re-parameterization has limitations such as the difficulty of removing residual connections across non-linear layers. This paper analyzes the limitation of re-parameterization method to remove residual connections such as RepVGG from both forward paths and backward paths.
To overcome the limitation, authors propose RM operation to equivalently remove residual connections from ResNet-based architectures which enables the plain models to keep higher performance for deeper networks. I think this ideology is very inspiring and may have an interesting combination with other research areas in the future.

3. I especially like the part of network pruning. It is known that ResNet is harder to be pruned compared to plain model because of the existence of residual connections. This paper proposes a alternative way to prune ResNet. First, use RM operation to convert ResNet to RMNet, then perform pruning on RMNet. The result shows the speed is much faster than vanilla pruning on ResNet architecture.

4. The part of the experiment for transforming MobileNetV2 is interesting. By further utilizing fusing operation, RM can transform MobileNetV2 into MobileNetV1 and the performance of MobileNetV1 keeps the same as MobileNetV2 which suggests that RM is applicable for light-weight models.

Weakness:

RM operation adds extra parameters and FLOPs for achieving equivalent transformation. Thus the benefit of the speed may be subject to the specific platforms, or additional operation like fusing two 1x1 Conv when converting MobileNetV2 into MobileNetV1.

**Summary Of The Paper:**

This paper focuses on an important direction of removing residual connections from ResNet-based architectures and proposes RM operation to achieve this goal. RM operation consists of two steps: reserving and merging. The reserving operation allows input feature map passing through the Conv, BN,ReLU layers without changing their values and the merging operation adds the output feature map and the reserved input feature map together with the last Conv layer. Unlike re-parameterization method in RepVGG, RM Operation can remove residual connection across non-linear layers, resulting in equivalent transformation from ResNet to plain models. Authors also design a series of plain models with RM Operation named RMNet, which outperforms previous plain models such as DiracNet, ResDistill and RepVGG.

**Summary Of The Review:**

The paper puts forward a novel method: RM Operation, which can equivalently remove residual connection across non-linear layer in ResNet-based architecture and shows great power in network pruning. I think RM operation is novel and can inspire future works on model design.

---

> ### Author Response · Authors · 2021-11-15
> **Response to  Reviewer oEx8**
>
> - Q1: RM operation adds extra parameters and FLOPs for achieving equivalent transformation. Thus the benefit of the speed may be subject to the specific platforms, or additional operations like fusing two 1x1 Conv when converting MobileNetV2 into MobileNetV1.
>
> - A1: Even though RM operation brings additional channels, the plain model is very friendly for deployment. Besides, we can utilize the advantage of the plain model to combine RM operation with other methods to further improve model performance, as shown in the paper.

---

> > ### Comment · Reviewer_oEx8 · 2021-11-29
> > **Response to Authors**
> >
> > After reading all the reviews and responses, I decide to keep my score. This paper does have some weaknesses such as bringing additional channels when converting ResNet to plain models. However, authors sufficiently demonstrate its effectiveness by combining pruning and fusion methods. Overall, the solution of equally converting ResNet to Plain models is very intersting and original in the literature. The results are also promising. I think it is worth being published and heard about in the community.

---

### Official Review · Reviewer_ybcv · 2021-11-02

**Correctness:** 3
**Technical Novelty And Significance:** 3
**Empirical Novelty And Significance:** 3
**Recommendation:** 6
**Confidence:** 5

**Details Of Ethics Concerns:**

No ethical concerns.

**Main Review:**

Main Review (+ = positive comment, o = neutral comment, - = negative comment)
------------------

\+ The strength of the paper is the relatively easy to implement method that can be applied to many ResNet variants in order to remove the residual connections. The results and discussion on finetuning, pruning etc. are also quite useful.

\+ The paper has good empirical part that covers quite well comparison to related work, results on fine-tuning and pruning.

o While the paper is somewhat incremental in nature, it generalizes the previously proposed approaches to removing residual connections to other ResNet architecture variants, so it has importance in that sense.

o While the paper is relative practical in nature, it is fine, since as far as I understand the resulting RMNet is mathematically equivalent to the corresponding ResNets, it inherits all analysis from ResNets.

\- Some details are missing in the paper. For example in section 3, e.g., Fig. 1 and the text describe the setup with the traditional ResNet with only one relu inside the block, while MobileNetV2 has a different architecture (resembling more preactivation resnet), where the residual branch has no nonlinearities. It is mentioned that PReLU is used in the case of MobileNetV2, but it not clear weather the parameters of PReLU fixed with some pattern (identity for new and relu for others?) or learned using SGD. The paper only mentions that the PReLU for the additional channels uses weights “set to one”.

\- Could the authors discuss whether it is possible to use fixed filters in part of the network and train the final RmNet from scratch? If I understand correctly, RmNet is mathematically equivalent to the corresponding ResNet and this should suggest that this is possible. This would remove the need for multi-step setup, where first there is a resnet trained, then the RM operation is applied, then the network is potentially fine-tuned and pruned. Since this paper is relatively practical work, it would be good to describe how to do this in typical deep learning framework as well.

\- Applicability to typical preactivation PreActResNet is missing (although there is MobileNetV2)

\- Applicability to some other popular ResNet variants, such as shufflenet is missing

Minor problems
------------------------------

\- Merging operation could be shown in pseudo-code in addition to the mathematical description, but this is a minor point, since there is also a figure about it.

\- Spelling mistakes, page 4. “(ie.e, in ResNet, every ResBlock has a following a ReLU layer, which keeps input values are all non-negative)”

\- Grammar error page 13 A.3. Finetune, the sentence “Thus We statistics the mean and variance by inputting ...” is wrong and sound probably by “Thus we computed the mean and variance by inputting…”


**Summary Of The Paper:**

Update: I have read the rebuttal, see below.
-----------
While residual connections are a key component in today's deep learning architectures, they can be problematic in some settings, e.g., in pruning. This paper presents an improved method (RMNet) for removing residual connections from a ResNet - type neural network after training. It improves over related work and in contrast to those, it works on some typical ResNet variants. The paper also discusses fine-tuning, pruning and more efficient architectures.

**Summary Of The Review:**

\+ Generalizes previous methods to other typical ResNet types

\+ Extensive Empirical evaluation

o/- Somewhat incremental, quite empirical

\- Some details missing

---

> ### Author Response · Authors · 2021-11-15
> **Response to Reviewer ybcv**
>
> - Q1: Some details are missing in the paper. For example in section 3, e.g., Fig. 1 and the text describe the setup with the traditional ResNet with only one ReLU inside the block, while MobileNetV2 has a different architecture (resembling more preactivation ResNet), where the residual branch has no nonlinearities. It is mentioned that PReLU is used in the case of MobileNetV2, but it is not clear whether the parameters of PReLU are fixed with some pattern (identity for new and ReLU for others?) or learned using SGD. The paper only mentions that the PReLU for the additional channels uses weights “set to one”.
>
> - A1: Firstly, we do not add any additional operation during training. RM operation is only applied for pre-trained MobileNetV2 when inference. It is worth noting that the position of ReLU is different in ResNet and MobileNetV2. In MobileNetV2, ReLU only exists inside the residual blocks. Thus the input feature map may have negative values and the ReLU will change their values. To preserve the feature map, we use PReLU to replace ReLU.  The parameters of PReLU regarding the additional channels are set to one. In this way, PReLU behaves as an Identity function.
>
> - Q2: Could the authors discuss whether it is possible to use fixed filters in part of the network and train the final RmNet from scratch? If I understand correctly, RmNet is mathematically equivalent to the corresponding ResNet and this should suggest that this is possible. This would remove the need for a multi-step setup, where first there is a ResNet trained, then the RM operation is applied, then the network is potentially fine-tuned and pruned. Since this paper is relatively practical work, it would be good to describe how to do this in a typical deep learning framework as well.
>
> - A2: Yes, we can initialize the additional channels of RMNet using Dirac initialization and fix their values. And, it is mathematically identical to training a ResNet and converting it to RMNet when inference. Besides, our method does not need re-training. Thus it is very convenient when one already has a pre-trained ResNet and wants to convert it to a plain model.
>
> - Q3: Applicability to typical preactivation PreActResNet and ShuffleNet is missing (although there is MobileNetV2)
>
> - A3: PreactResNet consists of [BN, ReLU, Conv, BN, ReLU, Conv]. We only need to insert a Dirac initialized Conv layer before the first BN layer, then we can use the same converting process as we perform on MobileNetV2: 1) using RM to remove the residual connection, 2) using fuse operation to merge two Conv layers.
>
> - Q4: Merging operation could be shown in pseudo-code in addition to the mathematical description, but this is a minor point since there is also a figure about it.- Spelling mistakes, page 4. “(ie., in ResNet, every ResBlock has a following a ReLU layer, which keeps input values are all non-negative)”- Grammar error page 13 A.3. Finetune, the sentence “Thus We statistics the mean and variance by inputting ...” is wrong and sound probable by “Thus we computed the mean and variance by inputting…”
>
> - A4: Thanks for the suggestion. We will carefully update our paper accordingly.
>
> - We will add all the mentioned code in the supplementary material (including MobileNetV2, PreActResNet, RMNet from Scratch, and SEBlock) and support more models in the future codebase.

---

### Official Review · Reviewer_Dt6G · 2021-11-03

**Correctness:** 3
**Technical Novelty And Significance:** 2
**Empirical Novelty And Significance:** 2
**Recommendation:** 6
**Confidence:** 2

**Main Review:**

The strength:
1. The approach has showed constant classification performance improvements on the benchmarks.
2. The idea is straightforward and easy to follow.

The weakness:
1. More experiments are needed to validate the performance, like the performance on detection, segmentation benchmarks.
2. Could the author list more inference comparison results across different hardware platforms, e.g,. cpu, gpu, popular embedded devices ?
3. Beside the RM steps, there are also other additional operations like pruning operations, it would be good to see the improvements from different operations.



**Summary Of The Paper:**

This work focuses on removing residual connection in network via  reserving and merging (RM) operations on the ResBlock. The author has tested the approach on classification tasks on several networks with skip connection, e.g., resnet, mobilenet v2, etc. The experiments demonstrated the good performance on the listed benchmarks.

**Summary Of The Review:**

In general, the paper has showed the strength in removing the residual connections on the classification tasks, it would be much more convincing to have more experiments on other popular tasks.

---

> ### Author Response · Authors · 2021-11-15
> **Response to Reviewer Dt6G**
>
> - Q1: More experiments are needed to validate the performance, like the performance on detection, segmentation benchmarks.
> - A1: We have performed experiments on detection and segmentation tasks. It can be observed from the following table, that RMNet is more superior to vanilla RepVGG on the vision tasks.
>
> | Method    | Boncbone     | Detection（mAP） | Insicent Segmentation （mAP） | Imgs/Sec |
> | --------- | ------------ | ---------------- | ----------------------------- | -------- |
> | Mask RCNN | RepVGG-B2    | 0.3620           | 0.3240                        | 10.9     |
> |           | **RMNet-B2** | **0.3830**       | **0.3440**                    | **10.9** |
>
> - Q2: Could the author list more inference comparison results across different hardware platforms, e.g., CPU, GPU, popular embedded devices?
> - A2: We compare RMNet with SOTA plain model (RepVGG) on different platforms. From the Table, it can be observed that RMNet achieves better speed-accuracy trade-offs, which implies that RMNet can be generally applied on variant platforms.
>
> | Arch              | Accuracy(%) | V100(Imgs/Sec) | 2080ti(Imgs/Sec) | CPU(Imgs/Sec) |
> | ----------------- | ----------- | -------------- | ---------------- | ------------- |
> | RepVGG B2         | 78.78       | 712            | 611              | 24.33         |
> | **RMNet 50×5 32** | **79.08**   | **849**        | **627**          | **32.75**     |
>
> - Q3: Besides the RM steps, there are also other additional operations like pruning operations, it would be good to see the improvements from different operations.
>
> - A3: Thanks for your suggestion. We have performed these experiments in the following table. From the results, we can see that when only applying RM on ResNet or MobileNet, it will lower down the speed. However, the benefit of RM is that the resulting plain model makes it very convenient for pruning and fusing, leading to a more efficient network structure.
>
> | Arch                              | Accuracy(%) | Imgs/Sec  |
> | --------------------------------- | ----------- | --------- |
> | ResNet18                          | 95.41       | 14939     |
> | RMNet18 (with RM, w/o Pruning)    | 95.41       | 10445     |
> | **RMNet18 (with RM and Pruning)** | **94.03**   | **46236** |
> | ResNet18 (with Pruning)           | 93.37       | 32256     |
>
> | Arch                            | Accuracy(%) | Img/Sec   |
> | ------------------------------- | ----------- | --------- |
> | MobileNetV2                     | 92.07       | 19119     |
> | MobileNetV2 (with RM, w/o Fuse) | 92.07       | 16990     |
> | **RMNetV1 (with RM and Fuse)**  | **92.07**   | **27720** |
> | MobileNetV1                     | 91.31       | 27720     |

---

### Official Review · Reviewer_BQPh · 2021-11-07

**Correctness:** 2
**Technical Novelty And Significance:** 2
**Empirical Novelty And Significance:** 2
**Recommendation:** 3
**Confidence:** 5

**Main Review:**

As the proposed conversion from residual network to plain network exactly preserves function, any throughput benefits are entirely determined by which form of operations is more efficient for the underlying combination of software library and compute hardware running the network.  Absent knowledge of such interactions, there does not appear to be any intrinsic reason for preferring the proposed form for rewriting residual layers.  In fact, the proposed approach seems to add computational work (more convolutional filters, Dirac initialized) for the sole purpose of taking advantage of fast implementations of convolutional layers.  Would a better approach simply be to write a custom fused low-level kernel for executing an entire residual block?  What form is actually better if one considers the optimal implementation possible for the underlying hardware (e.g., CPU, GPU, or TPU)?

A similar question arises for the pruning approach.  If the functional forms of the residual and plain networks are equivalent, then any pruning operations on one form should have an equivalent expression in the other.  Why is it necessary to first convert to a plain network, instead of simply considering pruning of residual connections in the original form?


**Summary Of The Paper:**

This paper proposes to convert residual network architectures to equivalent "plain" networks after training.  This is accomplished by augmenting convolutional layers with Dirac initialized filters (folding extraction of the residual signal into the conv layer), and modifying the subsequent Batch Normalization and ReLU layers (converting ReLU to PReLU) accordingly.  Motivations for doing this are to increase inference throughput, and allow additional pruning of the resulting plain network.


**Summary Of The Review:**

I am not convinced that the core idea of the paper -- converting residual networks to an equivalent plain network -- addresses any fundamental issue.  The argument for speed of one vs the other is empirical, and may merely depend on what is optimized in the underlying software libraries.  To make a case here, the paper should provide analysis in terms of achievable parallel efficiency by an optimal implementation.  Similarly, the argument made for pruning RMNet seems to be one of convenience rather than fundamental difference -- is there not an equivalent (though perhaps not off-the-shelf) approach in terms of of pruning components of the original network?

---

> ### Author Response · Authors · 2021-11-15
> **Response to Reviewer BQPh**
>
> - Q1: In fact, the proposed approach seems to add computational work (more convolutional filters, Dirac initialized) for the sole purpose of taking advantage of fast implementations of convolutional layers. Would a better approach simply be to write a custom fused low-level kernel for executing an entire residual block? As the proposed conversion from residual network to plain network exactly preserves function, any throughput benefits are entirely determined by which form of operations is more efficient for the underlying combination of software library and compute hardware running the network. Absent knowledge of such interactions, there does not appear to be any intrinsic reason for preferring the proposed form for rewriting residual layers.
>
> - A1: Firstly, we want to emphasize that there always exists costs to convert ResNet to plain models. For example, the model of RepVGG is at the cost of losing representation capability, especially for the deeper model, as we point out in the paper. Our proposed RM operation does bring additional channels for converting residual block to plain block. However, we achieve equivalent transformation which does not lose the representation capability of ResNet. In general, RM operation may lower down the speed. Thus we propose the following applications with RM operation: 1) Improving the performance of pruning by RM operation; 2) Merging two adjacent Conv blocks of MobileNetV2 into faster MobileNetV1 by RM operation; 3) Designing IBR block to decrease the number of additional channels brought by RM which achieves SOTA accuracy- speed-trade-off for plain models. We believe by combining RM operation with other methods, more superior plain models can be designed. Further, we observe that RM operation can help RepVGG break its depth limitation, improving performance to a higher level with the same inference structure.
> | Arch           | CIFAR-10 Acc(%) | CIFAR-100 Acc(%) | ImageNet Acc(%) |
> | -------------- | --------------- | ---------------- | --------------- |
> | RepVGG-133     | 84.1            | 46.64            | 70.91           |
> | **RMNet-133** | **94.75**       | **75.31**        | **74.56**       |
>
> - Q2: If the functional forms of the residual and plain networks are equivalent, then any pruning operations on one form should have an equivalent expression in the other. Why is it necessary to first convert to a plain network, instead of simply considering pruning of residual connections in the original form?
>
> - A2: The existence of residual connection requires that the number of channels of the input feature map is equal to the number of channels of the output feature map. Pruning one channel would affect all the corresponding input and output feature maps. Thus current pruning methods cannot prune all the invalid channels without affecting valid channels. Motivated by this phenomenon, we first convert ResNet into a plain model using RM operation, then perform efficient pruning. Since the plain model does not have residual connections, we can prune more parameters without hurting model performance. Besides, plain models are more friendly for deployment, resulting in faster inference speed.

---

> > ### Comment · Reviewer_BQPh · 2021-11-28
> > **residual connections do not constrain channel/filter pruning; fundamental question on speed unresolved**
> >
> > "Pruning one channel would affect all the corresponding input and output feature maps. Thus current pruning methods cannot prune all the invalid channels without affecting valid channels. Motivated by this phenomenon, we first convert ResNet into a plain model using RM operation, then perform efficient pruning. Since the plain model does not have residual connections, we can prune more parameters without hurting model performance."
> >
> > This claim is incorrect.  Consider the computational block around which a residual connection is made.  One can prune the output channels of this block so that it has fewer output channels than the residual connection.  Each of the block's output channels will match to one of the residual connection channels.  Some residual connection channels will have no matching channel from the block's output.  The former (matched channels) act as usual residual connections (input + block output), while the latter act as pass-through (input).  The only minimal overhead is an indexing operation that maps each block output channel to a corresponing residual connection channel (trivial in comparison to the computational work of layers within the block).  Similarly, one can prune block input channels: simply allow the block to operate on a subset of the full set of channels and use an appropriate indexing operation.  Again, this trivial overhead in comparison to, say, the computation of a convolutional layer within the block.  The presence of a residual connection does not constrain the ability to prune the block running parallel to that connection; layers within that block can be pruned in a structured (e.g., input, output channels) or unstructured manner or any combination thereof.
> >
> > "Plain models are more friendly for deployment, resulting in faster inference speed"
> >
> > The core question in my review was whether or not this observation was incidental: a consequence of which kernels currently have optimal low-level implementations in popular frameworks.  Hence, my comments asked for analysis at a more fundamental level -- how does the compute workload compare in terms of operation count for optimal implementations of each approach?
> >
> > I am not convinced the paper makes a compelling case here.  Section 4.2 states: "RMNet removes residual connections in the cost of bringing additional parameters. For example, in Figure 1, RM operation doubles the number of parameters of the original ResBlock. To alleviate this issue, we use RM operation on the ResNet-based architectures with Inverted Residual Block for designing RMNet."  Table 1 seems to lack an apples-to-apples comparison of a standard ResNet architecture and an RMNet of equivalent depth and block configuration (without any design other changes).

---

> > > ### Author Response · Authors · 2021-11-29
> > > **Response to Reviewer BQPh**
> > >
> > > - Q1: The claim about pruning is incorrect: index selection and unstructured pruning can be used for pruning ResNet.
> > > - A1: To avoid confusion, we did not mention index selection and unstructured pruning in our previous response. It is true that with index selection, one can prune the input/output channels of this block so that they have fewer channels than the residual connection. However, models with index selection have major deployment and inference issues while our main motivation is to design an efficient network that is friendly for inference and deployment.  As ShuffleNetV2 [R1] pointed out, Element-wise operations reduce the degree of parallelism. For example, the inference speed of ResNet50 will decrease by 25% if "index selection" is applied to each residual block. For ResNet50 with index selection, we need to prune 18.35% parameters to make the speed equal to vanilla ResNet50. Further, index selection and unstructured pruning are not supported by *torch.jit* and *TensorRT* which makes it harder to deploy on the general platform.
> > > It is worth noting that we use **structure pruning** on RMNet, resulting in a plain model without any residual connection and index, making it very easy to deploy on the platforms. We will further clarify this in the final submission.
> > > - Q2: Optimal low-level implementations for ResNet:
> > > -A2: First, Residual Distillation [R2] pointed out that the shortcuts in ResNet-50 account for about 40 percent of the entire memory usage on feature maps, and without residual connections, one can have 1.4× speed-up and 1.25× memory reduction. RepVGG [R3] also states that residual connections will reduce the degree of parallelism hence slowing down the inference. Thus it is commonly agreed that plain models are more superior to ResNet-based architecture.
> > > Secondly, designing optimal low-level implementation for different types of network architectures is indeed interesting and important but it is beyond our paper's scope, also the scope of other relevant papers that design plain models such as ResDistill, RepVGG [R2] [R3]. The main contribution of our paper is to propose RM operation to *equally* convert ResNet-based architecture to plain models. The speed testing among models is also performed in a very fair way (equal platform, equal batch-size, etc.), following the convention in the literature of designing plain models [R2][R3].  However, we would like to seek the solution of low-level implementation in the future.
> > > - Q3: Table 1 seems to lack an apples-to-apples comparison of a standard ResNet architecture and an RMNet of equivalent depth and block configuration.
> > > - A3: As our response to Reviewer Dt6G, It is true that directly applying RM operation on ResNet may lower down the speed. However, the real benefit of RM operation is the resulting plain model makes it very convenient for pruning and fusing, leading to a more efficient network structure. From the third Table in our response to Reviewer Dt6G, we can see RM operation can exhibit great power with pruning and fusing. We will add the Table in our final submission.
> > >
> > > | Arch                              | Accuracy(%) | Imgs/Sec  |
> > > | :-------------------------------- | :---------- | :-------- |
> > > | ResNet18                          | 95.41       | 14939     |
> > > | RMNet18 (with RM, w/o Pruning)    | 95.41       | 10445     |
> > > | **RMNet18 (with RM and Pruning)** | **94.03**   | **46236** |
> > > | ResNet18 (with Pruning)           | 93.37       | 32256     |
> > >
> > > | Arch                            | Accuracy(%) | Img/Sec   |
> > > | :------------------------------ | :---------- | :-------- |
> > > | MobileNetV2                     | 92.07       | 19119     |
> > > | MobileNetV2 (with RM, w/o Fuse) | 92.07       | 16990     |
> > > | **RMNetV1 (with RM and Fuse)**  | **92.07**   | **27720** |
> > > | MobileNetV1                     | 91.31       | 27720     |
> > >
> > > [R1]. [ShuffleNet V2: Practical Guidelines for Efficient CNN Architecture Design](https://arxiv.org/abs/1807.11164)
> > >
> > > [R2]. [Residual Distillation: Towards Portable Deep Neural Networks without Shortcuts](https://papers.nips.cc/paper/2020/hash/657b96f0592803e25a4f07166fff289a-Abstract.html)
> > >
> > > [R3]. [RepVGG: Making VGG-style ConvNets Great Again](https://arxiv.org/abs/2101.03697)

---

> ### Author Response · Authors · 2021-11-24
> **Response to Reviewer BQPh**
>
> Dear reviewer BQPh,
>
> Does our response address all of your concerns?  Please feel free to let us know if you have further questions.

---

> ### Author Response · Authors · 2021-12-01
> **Looking forward to your feedback (Part 1)**
>
> Dear reviewer BQPh,
>
> We are wondering if our response has addressed your concerns. Considering your previous questions, we first reply to your concerns in a better way, then make a short summary and contribution of our paper.
>
> - Q1: The benefits are entirely determined by which form of operations is more efficient for the underlying combination of a software library and computing hardware running the network. Would a better approach simply be to write a custom fused low-level kernel for executing an entire residual block? What form is actually better if one considers the optimal implementation possible for the underlying hardware?
>
> - A1: First, Residual Distillation [R1] pointed out that the shortcuts in ResNet-50 account for about 40% of the entire memory usage on feature maps, and without residual connections, one can have 1.4× speed-up and 1.25× memory reduction. RepVGG [R2] also states that residual connections will reduce the degree of parallelism hence slowing down the inference. Thus it is commonly agreed that plain models are more superior to ResNet-based architecture.     Secondly, designing optimal low-level implementation for different types of network architectures is indeed interesting and important but it is beyond our paper's scope, also the scope of other relevant papers that design plain models such as ResDistill, RepVGG [R1] [R2]. The speed testing among models is performed in a very fair way, following the convention in the literature of designing plain models.
>
> - Q2: The presence of residual connections does not constrain the ability to prune the block. If the functional forms of the residual and plain networks are equivalent, then any pruning operations on one form should have an equivalent expression in the other. Why is it necessary to first convert to a plain network, instead of simply considering pruning of residual connections in the original form?
>
> - A2: The existence of residual connection requires that the number of channels of the input feature map is equal to the number of channels of the output feature map. Pruning one channel would affect all the corresponding input and output feature maps, **which constrains the ability to prune**. Generally, there are three methods to deal with residual connection in pruning:
>
>   - **Only pruning the internal layers of the residual blocks [R3,R4,R5,R6].** However, this method cannot prune the invalid filter at input channels and output channels of ResBlocks.
>   - **Pruning all the corresponding input and output channels in all the blocks [R7,R8,R9,R10].** However, this method cannot prune all the invalid channels without affecting valid channels nor reserve all the valid channels while pruning all the invalid channels.
>   - **Pruning input or output channels with the help of index selection [R11,R12].** As ShuffleNetV2 [R13] pointed out, Element-wise operations reduce the degree of parallelism. Further, index selection and unstructured pruning are not supported by *torch.jit* and *TensorRT* which makes it harder to deploy on the general platform. For example, ResNet-50 with indexes will lower down the inference speed by 25%.
>
> From above, we want to highlight that even though the converted RMNet and ResNet have the same output, **the pruning operations do not have an equivalent expression because the existence of residual connection will limit the power of pruning in ResNet**. Thus, apart from the above three common approaches to prune ResNet, RM operation provides an alternative way, *i.e.,* equivalently converting ResNet into a plain model (RMNet) before performing any pruning method. By doing this, we can fully release the power of pruning. Figure 6 in our paper also verifies its effectiveness. We believe such insight may inspire future research in the pruning community.
>
> - Q3: Table 1 seems to lack an apples-to-apples comparison of a standard ResNet architecture and an RMNet of equivalent depth and block configuration.
>
> - A3: From the Table below ( also the third Table in our response to Reviewer Dt6G), we can see RM operation can exhibit great power with pruning and fusing when compared to standard ResNet architecture.
>
> | Arch                              | Accuracy(%) | Imgs/Sec  |
> | :-------------------------------- | :---------- | :-------- |
> | ResNet18                          | 95.41       | 14939     |
> | RMNet18 (with RM, w/o Pruning)    | 95.41       | 10445     |
> | **RMNet18 (with RM and Pruning)** | **94.03**   | **46236** |
> | ResNet18 (with Pruning)           | 93.37       | 32256     |
>
> | Arch                            | Accuracy(%) | Img/Sec   |
> | :------------------------------ | :---------- | :-------- |
> | MobileNetV2                     | 92.07       | 19119     |
> | MobileNetV2 (with RM, w/o Fuse) | 92.07       | 16990     |
> | **RMNetV1 (with RM and Fuse)**  | **92.07**   | **27720** |
> | MobileNetV1                     | 91.31       | 27720     |

---

> ### Author Response · Authors · 2021-12-01
> **Looking forward to your feedback (Part 2)**
>
>
> - Summary of the paper
>
> In this paper, we put forward an RM Operation for equivalent convert ResNet to VGG, MobileNetV2 to MobileNetV1.
>
> Although bringing additional channels, the equivalent transformation does not lose the representation capability of ResNet and MobileNetV2. And We propose the following applications with RM operation: 1) Improving the performance of pruning by RM operation; 2) Merging two adjacent Conv blocks of MobileNetV2 into faster MobileNetV1 by RM operation; 3) Designing IBR block to decrease the number of additional channels brought by RM which achieves SOTA accuracy- speed-trade-off for plain models. 4) Help RepVGG break its depth limitation, improving performance to a higher level with the **same inference structure** (see table below).
>
>   | Arch          | CIFAR-10 Acc(%) | CIFAR-100 Acc(%) | ImageNet Acc(%) |
>   | ------------- | --------------- | ---------------- | --------------- |
>   | RepVGG-21     | 94.88           | 76.43            | 72.5            |
>   | **RMNet-21**  | **95.11**       | **76.96**        | **72.58**       |
>   | RepVGG-69     | 93.42           | 74.45            | 74.52           |
>   | **RMNet-69**  | **94.73**       | **76.72**        | **75.08**       |
>   | RepVGG-133    | 84.1            | 46.64            | 70.91           |
>   | **RMNet-133** | **94.75**       | **75.31**        | **74.56**       |
>
> We believe by combining RM operation with other methods, more superior plain models can be designed.
> Please feel free to let us know if you have further concerns.
>
> [R1]. [Residual Distillation: Towards Portable Deep Neural Networks without Shortcuts](https://papers.nips.cc/paper/2020/hash/657b96f0592803e25a4f07166fff289a-Abstract.html)
>
> [R2]. [RepVGG: Making VGG-style ConvNets Great Again](https://arxiv.org/abs/2101.03697)
>
> [R3]. [ThiNet: A Filter Level Pruning Method for Deep Neural Network Compression](https://openaccess.thecvf.com/content_iccv_2017/html/Luo_ThiNet_A_Filter_ICCV_2017_paper.html)
>
> [R4]. [Data-Driven Sparse Structure Selection for Deep Neural Networks](https://openaccess.thecvf.com/content_ECCV_2018/html/Zehao_Huang_Data-Driven_Sparse_Structure_ECCV_2018_paper.html)
>
> [R5]. [Discrimination-aware Channel Pruning for Deep Neural Networks](https://arxiv.org/abs/1810.11809)
>
> [R6]. [Neuron Merging: Compensating for Pruned Neurons](https://arxiv.org/abs/2010.13160)
>
> [R7]. [Pruning Filters for Efficient ConvNets](https://arxiv.org/abs/1608.08710)
>
> [R8]. [Gate Decorator: Global Filter Pruning Method for Accelerating Deep Convolutional Neural Networks](https://arxiv.org/abs/1909.08174)
>
> [R9]. [Neural Network Pruning with Residual-Connections and Limited-Data](https://openaccess.thecvf.com/content_CVPR_2020/html/Luo_Neural_Network_Pruning_With_Residual-Connections_and_Limited-Data_CVPR_2020_paper.html)
>
> [R10]. [Group Fisher Pruning for Practical Network Compression](https://proceedings.mlr.press/v139/liu21ab.html)
>
> [R11]. [Learning Efficient Convolutional Networks through Network Slimming](https://openaccess.thecvf.com/content_iccv_2017/html/Liu_Learning_Efficient_Convolutional_ICCV_2017_paper.html)
>
> [R12]. [Channel Pruning for Accelerating Very Deep Neural Networks](https://openaccess.thecvf.com/content_iccv_2017/html/He_Channel_Pruning_for_ICCV_2017_paper.html)
>
> [R13]. [ShuffleNet V2: Practical Guidelines for Efficient CNN Architecture Design](https://arxiv.org/abs/1807.11164)

---

### Decision · Program_Chairs · 2022-01-20

**Decision:**

Reject

**Comment:**

This paper propose a reparametrization approach for pruning residual networks. The proposed approach replace the skip layer connections with feedforward layers, and show the equivalence to the original network. However, the current presentation is not very clear on the advantage of the proposed approach for pruning. As two networks are equivalent, pruning the reparameterized network can be transferred to pruning the residual network. The authors need to clarify how their reparametrized network is different from the residual network when being pruned. More ablation studies are also need to better justify their claim.